# A Qualitative Study on the Barriers and Enablers to Effective Hypertension Management in Ghana

**DOI:** 10.3390/healthcare13050479

**Published:** 2025-02-22

**Authors:** Francis Sambah, Kristin McBain-Rigg, Abdul-Aziz Seidu, Theophilus I. Emeto

**Affiliations:** 1Public Health and Tropical Medicine, James Cook University, Townsville, QLD 4811, Australia; kris.mcbain@jcu.edu.au (K.M.-R.); abdulaziz.seidu@jcu.edu.au (A.-A.S.); 2Department of Health, Physical Education and Recreation, University of Cape Coast, Cape Coast P.O. Box UC 182, Ghana; 3World Health Organization Collaborating Center for Vector-Borne and Neglected Tropical Diseases, James Cook University, Townsville, QLD 4811, Australia

**Keywords:** patients, hypertension, management, enablers, barriers, Ghana

## Abstract

**Background:** Hypertension remains a significant public health challenge in Ghana. Understanding the experiences of hypertensive patients can inform strategies to improve their management. This study explored the perceived enablers and barriers to hypertension management among patients in the Ashanti region, Ghana, using the Chronic Care Model as a framework. **Methods:** In-depth interviews were conducted with 20 hypertensive patients receiving care at Komfo Anokye Teaching Hospital. Inductive thematic analysis was employed to identify key themes and subthemes. **Results:** Several barriers to hypertension management emerged, including economic constraints, environmental and lifestyle factors, knowledge and awareness deficits, medication-related issues, and policy and provider-level barriers. Conversely, enablers such as patient empowerment, education, healthcare access, and policy and provider support and relationships were identified. **Conclusions:** Effective hypertension management requires addressing a complex interplay of barriers and enablers. Interventions targeting economic factors, lifestyle modifications, knowledge dissemination, medication adherence, and systemic improvements are crucial. Additionally, empowering patients, enhancing education, improving healthcare access, and fostering strong provider–patient relationships can significantly contribute to better outcomes. Future research should investigate the impact of a multi-level intervention on hypertension management in Ghana.

## 1. Introduction

Building national capacity to combat non-communicable diseases (NCDs) is important, particularly in low- and-middle-income countries (LMICs) [1]. Globally, the prevalence of hypertension (HPT), a primary risk factor for many NCDs, continues to increase [2]. Epidemiological data reveal that HPT remains a significant health concern, with global prevalence ranging from 13% to 41% [2], 54% across Africa [3,4], 27% in Ghana [5], and 30.7% in Ghana’s Ashanti Region [A/R]. These figures indicate that HPT substantially contributes to mortality both locally [6] and globally [7]. The response of health systems to managing NCDs has been inconsistent, resulting in low rates of HPT control both locally [8] and across LMICs [9]. Current control trends threaten to undermine global targets aimed at reducing uncontrolled HPT from 25 to 30% by 2030 [10,11] and jeopardise the Sustainable Development Goal [SDG] 3.4, which seeks to decrease premature deaths from NCDs by one-third by 2030 [12]. This situation necessitates sustained efforts from a diverse array of stakeholders, including HPT patients, healthcare providers, government, and health systems, to effectively tackle the inherent challenges.

Despite the considerable global burden of HPT, awareness, and treatment remains inadequate. Only 34% of Africans recognise their hypertensive status, 31.3% are receiving medication, and 6.5% believe their HPT is under control [13]. This deficiency raises concerns among stakeholders [10]. Effective health programmes cannot be implemented without the engagement of a broad spectrum of stakeholders [14], including patients, policymakers, and healthcare practitioners, who must collaborate continuously to manage conditions such as HPT [14]. However, productive provider–patient interactions are often hampered by both endogenous and exogenous factors, adversely affecting patient health outcomes. Thus, understanding the perspectives of individuals with HPT is important for applying available evidence to disease management [15]. Patients’ self-management practices can reveal critical bottlenecks within the care process, thereby enhancing future quality of care.

Numerous studies have identified factors enabling effective HPT management [16,17,18,19,20]. For instance, research conducted in Nepal identified family support, positive illness perceptions, free healthcare services, patient knowledge about HPT, medication reminders, proximity to health facilities, perceived severity of the condition, availability of healthcare services, and physician recommendations as significant enablers to HPT control [16,17]. A study by Nakwafila et al. [19] in Namibia highlighted patient education, drug availability, and support systems as essential facilitators to HPT management. In Ghana, Byiringiro et al. [20] noted that the presence of wellness and HPT clinics at the primary healthcare level, patients owning personal blood pressure (BP) monitors, and nurses’ expanded roles in HPT control efforts served as key enablers. While these facilitators present valuable opportunities for HPT management, the context-specific nature of the findings may limit their broader applicability, particularly in relation to context-specific barriers that could undermine these potential gains.

Barriers to effective HPT management are diverse, encompassing provider, healthcare system, and patient-level challenges that significantly hinder successful control efforts [10,19,20,21,22]. Systematic reviews highlight patient- related challenges in self-management of HPT [23,24]. In Nepal, key barriers include limited HPT knowledge and its treatment, poor adherence to antihypertensive medication, inconsistent follow-up by patients, lack of standard treatment protocols, and inadequate health education provided by healthcare professionals [25,26]. Similarly, a Namibian study revealed that knowledge gaps, insufficient social support, the complexity of multiple drug regimens, side effects of medications, and drug shortages significantly hinder patients’ self-management efforts [19]. Furthermore, Galson et al. [27] in Tanzania identified a range of barriers affecting HPT control, including patients’ knowledge gaps, disease severity, communication with healthcare providers, follow-up care, family support, medication costs, access to health services, waiting times, transportation costs, and fear.

In Ghana, previous studies have highlighted a variety of barriers impeding effective HPT management [20,28,29]. These include patient-level issues such as negative self-management behaviours, socio-economic factors, medication non-adherence, reliance on traditional treatments, and sociocultural influences. At the healthcare provider level, barriers include perceived low quality of antihypertensive medications, communication gaps, inadequate collaboration among healthcare teams, insufficient training on HPT, and challenges related to patient referrals. Additionally, systemic barriers encompass a lack of investment and funding, inadequate health facilities, shortages of antihypertensive medications and healthcare providers, logistical challenges, and issues related to national health insurance and policy frameworks [20,28,29]. If these barriers are not thoroughly examined and addressed, they may significantly undermine empirical interventions aimed at managing uncontrolled HPT in Ghana.

Several studies in Ghana have explored the perspectives of various stakeholders regarding the enablers [20] and barriers [20,28,29] to effectively manage HPT. However, many of these investigations have primarily focused on healthcare providers [20,29], while Laar et al. [28] included HPT patients as part of their participant pool, albeit in a peri-urban setting in the Eastern region of Ghana. The current study aims to examine HPT management at the Komfo Anokye Teaching Hospital (KATH), a major tertiary referral facility serving eight administrative regions in Ghana. This study is grounded within the Chronic Care Model (CCM) [14] to systematically unpack the barriers and enablers impacting HPT management, thereby aligning with national and global objectives of achieving a 25% reduction in HPT prevalence by 2025 [10].

### Conceptual Framework

This study is part of a larger study that utilises CCM [14] as its theoretical framework. The primary objective is to assess the perspectives of patients with HPT regarding their perceived enablers and barriers to effective management of their condition in Ghana. The model posits that health outcomes for patients result from the collective actions and inputs of various stakeholders, including governmental health policies and investments, as well as healthcare providers who implement clinical protocols and practices in managing chronic conditions, such as HPT [14]. By integrating the roles of healthcare practitioners and patients, the model seeks to enhance health outcomes [30]. The major components of the model are six independent constructs [14]. Four (4) of the constructs, “health system organization”, “delivery system support”, “decision support”, and “clinical information support”, are concerned with the practices of healthcare providers, whereas the other two, “self-management” and “community resources and policies”, concerns the patients [31]. In this study, “self-management” refers to the actions and strategies patients adopt to control their blood pressure, reduce complications, and improve their overall well-being [14,32]. It encompasses both pharmacological and non-pharmacological strategies essential for effective control of HPT [33]. The “self-management support” in the model was adapted for this study to imply HPT patients’ disease management practices, which have been found to be limited by barriers such as lack of medication adherence, poor knowledge of antihypertensive issues, and lifestyle modification challenges [10]. The present study, therefore, evaluated these patient-level barriers and their potential to hinder Ghana’s progress towards HPT control. It also sought to unpack enablers to mitigate barriers affecting HPT patients’ “self-management” practices with effectual improved health outcomes. Consequently, this study explored the experiences of HPT patients, focusing on the enablers and barriers that influence their management efforts. The CCM is particularly appropriate for this investigation, as it facilitates the assessment of the self-management practices of HPT patients to unpack the barriers and enablers impacting HPT management. Moreover, this model has been widely validated in similar studies internationally [30,34,35] and applied within the Ghanaian context [36].

## 2. Materials and Methods

### 2.1. Ethics Approval

This study adhered to the ethical principles outlined in the Declaration of Helsinki [37]. Ethical approval was obtained from the James Cook University (JCU) Human Ethics Committee (H9031), the KATH Ethical Review Committee (KATHIRB/AP/029/23), and the Ghana Health Service (GHS) Ethics Review Committee (GHS-ERC: 005/09/22). Additional approval was granted by the Ashanti Regional Health Directorate. Informed consent, both written and verbal, was obtained from participants after a detailed explanation of the study’s objectives, procedures, potential risks, and benefits. Participants were assured of confidentiality and the right to withdraw at any time.

### 2.2. Study Setting and Design

This study was conducted at KATH in the Ashanti Region (A/R) of Ghana. KATH is one of the major referral hospitals serving a large portion of the Ghanaian population. The A/R was selected due to its high prevalence of HPT, with data showing 15.3% of women and 18.1% of men between 15 and 49 years with HPT, the highest in Ghana [38]. KATH is also famous for its pool of skilled healthcare providers on HPT patients’ management [39] and provides an opportunity to access HPT patients with varying degrees of severity, allowing the researchers to have a comprehensive assessment of the enablers and barriers affecting HPT management.

This study is part of a larger project that evaluated Ghana’s progress towards HPT control in the A/R. The study seeks to address the following objectives: To assess the extent of Ghana’s adherence to the PASCAR’s 10-point action plan towards HPT control; to assess patient barriers affecting HPT control in Ghana; to evaluate government and health system barriers to HPT control in Ghana; and to explore stakeholders’ perspectives on barriers and enablers to HPT control. However, this study used a descriptive qualitative design [40] and an inductive thematic analysis framework [41] to explore HPT patients’ perceived enablers and barriers to HPT management. This design allows for a detailed examination of the participants’ experiences, perceptions, and challenges in managing their condition and thus captures their contextual insights. The study followed the Consolidated Criteria for Reporting Qualitative Studies (COREQ) guidelines [42].

### 2.3. Target Population and Sample

The participants comprised a diverse group of adults diagnosed with HPT, varying in age, gender, socio-economic status, and educational background. This diversity ensured a comprehensive understanding of the factors influencing HPT management. A total of 20 HPT patients were recruited and selected conveniently during a pre-interview survey. This sample size was arrived at guided by the principle of data saturation, where additional interviews/data collection efforts no longer reveal new information [43]. The adequacy of this sample size aligns with the existing literature in qualitative studies in health [44,45] and, specifically, chronic disease management research [45] that demonstrates that data saturation could be achieved with 12 interviews in a homogeneous population [44] and between 10 and 20 participants when methodological rigour is maintained. This ensured our study was complete in terms of experiences and perspectives and unconstrained by arbitrary sample size, thereby lending more credibility and rigour to the findings.

Participants included HPT patients registered with the KATH HPT clinic who had attended follow-up appointments for at least one year and had previously participated in the earlier quantitative data collection process as part of the broader project. Additionally, HPT patients who were admitted but had taken part in the earlier quantitative data collection process, were alert, and provided consent were included in the study. Including only those who had engaged in the research earlier ensured that the participants indeed had relevant medical histories and experiences with HPT management at KATH, which in turn made their experiences more reliably comparable. HPT patients who were not registered members of the KATH HPT clinic, who did not participate in the earlier quantitative study phase, and who did not consent were excluded. This approach maintained consistency and continuity in data collection. Furthermore, excluding those who did not consent upheld ethical standards and respect for participants’ autonomy.

### 2.4. Data Collection

The interview guide was developed based on the findings of the quantitative data developed from the previous literature [10]. The first author (F.S) and the research assistant (RA) piloted the interview guide with five HPT patients who were excluded from the main study. No amendments were deemed necessary for the instrument. The interview guide has been attached as a supplementary (Appendix A). All in-depth interviews were conducted face-to-face in English and Twi by the trained RA and FS at various locations, including participants’ homes, the KATH HPT clinic, hospital wards, and other convenient sites for both participants and researchers. With participant consent, each interview was digitally recorded. Probes were employed as needed to clarify concepts and obtain detailed information. Field notes documented non-verbal cues and researcher reflections. The average duration of the interviews was approximately 40 min.

### 2.5. Data Analysis

Following the interview process, the Twi-language interviews were translated and transcribed into English by the RA, who is a native Twi speaker and has conducted similar cross-language qualitative research at both undergraduate and Master of Philosophy levels [46]. The first author (F.S.) cross-referenced these transcripts with the original audio recordings to ensure accuracy. Data analysis was conducted using NVivo version 12 (QSR International, Ltd., Daresbury Cheshire, UK) through an inductive thematic framework [41,47,48]. The inductive thematic analysis framework gave the researchers a chance to derive patterns and themes directly from the data, unconstrained by preexisting theories. Such an approach has been especially useful in trying to understand complex health behaviours and system-related factors that ensure findings are grounded in the actual experiences of participants and not theoretical constructs.

The transcripts were meticulously examined to identify key themes and points, with significant elements highlighted by the first author (F.S.). The research team met regularly to discuss divergent interpretations of data. Collaborative discussions with co-authors K.M.-R., A.S., and T.I.E. facilitated the refinement of the analysis framework and thematic categorisation. The first author initiated the grouping of codes into overarching themes, which were subsequently categorised into thematic networks by the research team. The research team, F.S., K.M.-R., A.S., and T.I.E., checked the themes for coherence to ensure that each theme described the main patterns of data. This process involved further analysis to explore relationships among the themes, leading to the identification of pertinent quotes that exemplified the themes and supported the findings. Where disagreement occurred, this was worked through by dialog, returning to the raw data, and ensuring that the accounts presented were consistent with participants’ narratives. Where necessary, K.M.-R., who is an expert in qualitative health research, was consulted to offer an impartial perspective. Sub-themes were generated where necessary to capture the subtleties of participant experiences. The final themes were checked against the dataset to ensure the findings were adequately and accurately represented.

The team collectively agreed upon the selection of quotes, ensuring they accurately represented the themes derived from the data. Ultimately, the final themes were substantiated with verbatim statements, augmented by relevant demographic details of participants, including sex, age, and HPT status. 

### 2.6. Trustworthiness and Reflexivity

Strategies delineated in establishing trustworthiness were adopted [49]. These included credibility, dependability, confirmability, and transferability [50]. The researchers employed varied strategies to ensure credibility. One of which was an extended engagement with the participants over a period of time in the field, offering us an opportunity to engage the participants in their HPT condition, from the patients’ self-management practices to social and environmental conditions impacting their HPT management, granting the team a comprehensive insight [51]. This extended engagement gave us the opportunity to build trust and rapport with the participants over time, giving us a chance to gain nuanced insights into their beliefs, experiences, and their HPT self-management behaviours [52]. Bracketing techniques were used in planning, conducting, and reporting this research to identify and address personal ideological assumptions held by the first author and potential influences on this study [53]. The research team was in constant contact throughout all stages to maintain transparency and critically examine the effect of any such assumptions. This brought clarity during data collection, and its analysis improved the quality of decision-making and finally strengthened the reliability of the findings [54]. Dependability was ensured through the study’s detailed protocol, ensuring an audit trail of all data collection procedures and precise coding. Coding was independently checked for consistency by the research team members [55]. Confirmability was assured through daily debriefing among the research team [52]. Engaging a healthcare qualitative researcher who exerts supervisory checks to review interpretations and findings minimises researcher bias, and obtaining feedback from the supervisory team helps to validate the interpretations and minimises personal biases brought alternative perspectives [52]. The study ensured transferability through an elaborate description of its methodology, which enables other researchers to replicate it. It adequately described the setting where the study was carried out, the criteria for including participants, and its convenient sampling approach. These methodological details make the study more transferable, as they guarantee its use in other settings [52].

The study also adopted reflexivity strategies to help minimise the influence of researcher assumptions, biases, and influences in the research process and findings [56]. The processes adopted consist of a reflexive journal, team debriefings, critical self-analysis, participant validation, seeking an inclusive point of view, and ethical reflexivity. On keeping a reflexive journal, the research team logged our feelings, thoughts, and decisions, which were discussed in an open manner to address any personal or collaborative biases [57]. Daily team debriefing sessions were held not only to discuss operational issues but also to reflect logs of the research team regarding interaction with participants and how they influenced the study, both real and assumed. Practices such as these heed recommendations by Berger [58] towards developing collaborative reflexivity among researchers. Also, ethical reflexivity was achieved in conducting and observing international ethical frameworks where participants’ autonomy and sensitivities were guaranteed [59]. Finally, seeking inclusive points of view through verification of themes and subthemes with the team tends to triangulate multiple viewpoints to eliminate individual researcher bias.

## 3. Results

### 3.1. Demographic Characteristics of Participants

The study included 20 participants, with 11 females and 9 males. Participants’ ages ranged from 48 to 89 years. The majority identified as Christian (n = 18) and Akans (n = 16). Twelve participants were married, and twelve had no formal education. The average duration of HPT was 8.5 years.

### 3.2. Themes Generated from the Data

An inductive thematic analysis revealed five major themes regarding barriers to patients’ HPT management: (a) economic burden, (b) environmental and lifestyle influences, (c) knowledge and awareness barriers, (d) medication-related and adherence barriers, and (e) policy and provider-level barriers. In contrast, four themes emerged as enablers for effective hypertension management: (a) acceptance and empowerment, (b) educational empowerment, (c) healthcare access and policy, and (d) provider support and relations. Figure 1 summarises the themes and subthemes of the study. Each theme is discussed in detail, supplemented with illustrative quotes from the participants.

### 3.3. Barriers to Hypertension Management

#### 3.3.1. Theme One: Economic Burden

The first theme identified from the data concerning barriers to HPT management was the economic burden. This theme encompasses two sub-themes: direct costs and insurance limitations. Participants shared their experiences with the healthcare system and their personal management of HPT, highlighting significant challenges related to the cost of antihypertensive medication and gaps in the health insurance system. Most participants indicated that the economic burden severely hampers efforts to manage HPT effectively.

##### Sub-Theme: Direct Cost

Many participants expressed concerns about the out-of-pocket payments required at each visit to the HPT clinic, noting that these costs continue to rise and act as a barrier to care. Some patients reported being referred to private pharmacies to purchase antihypertensive medications at exorbitant prices. Additionally, the unpredictable costs of health services often compel some patients to seek financial assistance from others, and those unable to obtain help sometimes leave without refilling their prescriptions.

“*We are in difficult times, but our health should matter most to government and the health system actors. Like the charges they unilaterally impose on us is in bad taste to our hypertension management progress. For my last two visits, they have consistently increased the charges which is a great disservice to the poor patients. So many patients who are not able to borrow from others had to go home without medication. I feel so bad that patients with chronic conditions like hypertension and on health insurance should be subjected to such pathetic treatment*”[Female HPT patient, 70 years].

##### Sub-Theme: Insurance Limitations

Participants also discussed challenges related to the national health insurance scheme. While some acknowledged that health insurance has been beneficial in reducing healthcare costs, the majority lamented that the current state of the health insurance system does not meet their expectations. Many decried the persistent out-of-pocket payments despite having insurance and the continuous increase in charges at HPT clinics, which act as a disincentive to effective HPT management. Some patients now perceive the health insurance system as ineffective and a departure from its initial promise to alleviate healthcare costs for subscribers.

“*At first it was good, now we pay money (GHS40.00) for every review which is not good. Some people leave without taking the drugs because they don’t have money to pay. I will say the policies are not effective. Why because the health insurance is basically losing it relevance. You subscribe for health insurance yet when you are sick and come to the hospital, if you don’t have money, you cannot still attend for health service. What is the essence of a policy you subscribe to yet cannot benefit from when the need arises. It’s a hoax insurance*”[Female HPT patient, 58 years].

#### 3.3.2. Theme Two: Environmental and Lifestyle Influences

Participants highlighted the pervasive influence of environmental factors, including the farmer’s use of chemical fertilisers and herbicides, and how these contribute to NCDs such as HPT. They expressed concern that these factors severely hinder patients’ ability to manage their HPT effectively.

##### Sub-Theme: Diet and Environmental Factors

Many participants reported that their dietary habits negatively impacted their ability to manage HPT. Some attributed their challenges in achieving successful HPT management to their diet, while others blamed illegal mining activities (locally referred to as galamsey), which introduce harmful chemical residues into the soil and subsequently into food crops. This perception underlines the greater implications of environmental degradation for public health. It is not only the compromised quality of diets due to toxic substances in food sources but also the increased risk of hypertension-related complications. Thus, effective management of hypertension requires not only personal adjustment in the diet but also systemic interventions in the form of regulating environmental hazards that impact food safety. Additionally, some participants admitted to neglecting lifestyle modifications, such as engaging in regular physical activity, which is a non-pharmacological measure essential for HPT management.

“*The food we eat now has brought about all this illness. At first there was nothing like chemicals but now is coming and even farmers apply right from planting, so it has affected our foods which has brought all these sicknesses to the world. Worst is the rampant galamsey activities in our farming areas with hazardous chemicals that seep into water bodies use for vegetables growing, irrigation and a whole lot. Those are part of our challenge and the reason for the increasing noncommunicable diseases in Ghana…Duty bearers are failing to implement and enforce our public health laws in the agricultural sector, the mining sector and the health sector as well on ineffective low-quality drugs, the rampant sugary and fast foods in our schools, it is just bad*”[Male HPT patient, 66 years].

##### Sub-Theme: Sociocultural Influence

Some participants indicated that their struggles with managing HPT were influenced by cultural beliefs that favour herbal medicine as a treatment alternative. They expressed that resorting to herbal remedies is a deeply ingrained practice within their culture. However, it was evident from their accounts that even those who rely on herbal medicine often revert to conventional antihypertensive treatments when blood pressure levels rise.

“*If I fail to take it, then is deliberate because I take my herbal drugs and don’t drink both together. For me, herbal medicine is part of my family heritage and our people tradition. Before orthodox medicine our herbal medicines were our medications, and we can’t ditch them like that. I can even tell you most of these orthodox drugs are extracts from herbal so why should we give away our herbs. I can attest that the herbal medicines I take is contributing hugely to the effective control of my condition. I usually give myself sometime to take the herbal drugs and later revert to the orthodox drugs when…I see my BP level rising before I take the orthodox drugs. These herbal drugs I use are not prescribed by any physician but sometimes I listen to specialist in herbal medicine and apply their directives together with the knowledge I gain from our family herbalist. I prepare the drug myself using the herbs the herbal experts teach, and it is giving me positive outcome*”[Male HPT patient, 66 years].

#### 3.3.3. Theme Three: Knowledge and Awareness Barriers

Another theme that emerged from the data is the lack of knowledge and awareness among patients regarding the determinants of HPT, which leads to delayed diagnosis and refusal to accept their diagnosis. This theme generated two sub-themes: lack of awareness regarding diagnosis and lack of educational empowerment. Patients displayed a deficiency in knowledge about HPT-related risk factors, signs, and symptoms that ultimately affect their ability to control the condition. Furthermore, the lack of educational opportunities to help them understand how to prevent or manage HPT results in delayed diagnoses and subsequent complications.

##### Subtheme: Lack of Awareness and Diagnosis

Patients reported a significant lack of knowledge and awareness of the precursors to HPT, which has contributed to detrimental health behaviours and delayed diagnosis. Many expressed frustrations over the late diagnosis of their condition, which adversely affected their management efforts. This reflects the bigger problem: lack of proper awareness and, consequently, late diagnosis that has impeded the effective control of HPT and, hence, its complications among the patients. So, once people lack knowledge about the early signs and symptoms relative to HPT, they may resort to unorthodox modes of managing their ill health, leading to late diagnosis. This is evident in the submission of one of the participants:

“*I didn’t know from the onset it was hypertension. I was experiencing throat dryness so after further consultation with the Doctor that I realized it was also a symptom of the condition. Because they couldn’t detect the condition from the previous clinics, I was taking anti-anaemic drugs which has high sugar content so after the diagnosis I was restricted from those drugs then the BP level started coming down*”[Female HPT patient, 70 years].

##### Subtheme: Lack of Educational Empowerment

Participants discussed how a lack of formal education or literacy hampers their ability to monitor their health and engage effectively with healthcare services. The participant’s account revealed the influence of low formal education on effective HPT management. Illiteracy prevents patients from reading about their disease, checking the medicines prescribed for them, and even remembering their next appointments. The participant confessed that education is important in helping one to identify possible mistakes in their care and to make appropriate decisions concerning their health. Without this knowledge, many patients like him have to depend on others for help, increasing their susceptibility to medication errors and mismanagement. This shows the need for patient education programmes so as to improve self-management and increase health outcomes. This was highlighted by one of the HPT patients as described hereto.

“*Education is good because you can learn more about the disease on your own. If they give you wrong medication you can easily detect but we those who lack the formal education, I sometimes have to ask people to check my hospital card for me to know my follow-up date. It is only God we depend upon. There was a time a pharmacy gave someone medicine that was not for him. It took an educated person to read before informing him the drug is not for his sickness so I will say education helps a lot. So, to answer your question, my lack of education is indeed affecting my adequate management of my condition, but it is what it is*”[Male HPT patient, 71 years].

Another participant acknowledged the impact of educational deficits on health literacy and how it could hinder effective HPT management.

“*I can’t read but I know the alphabet so if I have the monitoring apparatus, I will engage someone to assist me. I don’t know if is affecting my treatment of this condition. Only the doctors can tell. However, I think I would have better handled my hypertension situation if I could read and search for information concerning the disease myself. To an extent I can say my educational background is affecting my ability to self-manage my condition*”[Female HPT patient, 67 years].

#### 3.3.4. Theme Four: Medication-Related and Adherence Barriers

This theme elucidates the medication-related and adherence challenges faced by participants in managing their hypertensive condition. Two subthemes emerged: adherence challenges and medication shortages. This theme highlights the pharmacological deficiencies in the management of patients’ conditions. Participants frequently reported difficulties in obtaining adequate antihypertensive refills from government hospital pharmacies during routine follow-ups, primarily due to medication shortages. Consequently, many patients are directed to procure medications from private pharmacies, a process not covered by their health insurance. The additional financial burden often hampers their ability to adhere to prescribed regimens. Furthermore, some participants cited adverse effects associated with the antihypertensive medications provided by their health insurance as a contributing factor to their non-adherence.

##### Subtheme: Adherence Challenges

Participants expressed concerns regarding the side effects of antihypertensive medications, particularly those dispensed through their health insurance, which hindered their adherence to treatment protocols. Reported side effects included decreased libido and frequent urination, which disrupted sleep patterns and limited participation in sociocultural activities. One participant remarked the following:

“*I only buy the orthodox one the doctor prescribes for me. Because of the excessive urination that comes after taking the drugs so anytime I am traveling or going to any public gathering like going to the mosque, I don’t usually take it. This distort the fixed time I apply in the taking of the drug. So, I don’t have an exact time like 8 am or 9 am that I take my medication but the convenient time I am able to eat or available at home. I know it is a problem in the management of my condition, but my choices are limited*”[Female HPT patient, 67 years].

Other participants noted forgetfulness as a significant barrier, while some felt that their stable blood pressure levels justified skipping doses.

“*Well, forgetfulness is one of my biggest barriers as I have been skipping the time of my medication and a number of times, not able to take it at all. This has made me take my medication at different times against the advice of the doctors and pharmacies. This may be part of my struggle to manage my hypertension*”[Male HPT patient, 71 years].

##### Subtheme: Medication Shortages

Most participants reported frequent shortages of antihypertensive medications at hospital pharmacies during their visits to the hypertension clinic for refills. Consequently, patients are often issued prescriptions to obtain medications from private pharmacies, which imposes additional financial costs that many cannot afford, exacerbating their adherence challenges.

“*There is another issue of antihypertension medication shortage at the hospital. How can that be when government is aware of a chronic condition like hypertension requires patients to live on drugs. This situation leads to extra healthcare expenditure on the patient since we have to purchase the drugs from private pharmacies at exorbitant prices*”[Male HPT patient, 59 years].

#### 3.3.5. Theme Five: Policy and Provider Level Barriers

Another significant barrier identified in the data pertains to healthcare policy and provider-related gaps. Two subthemes emerged: policy and provider gaps and governmental lack of accountability, both of which impact patients’ management of HPT. Several participants expressed dissatisfaction with the current health policies, particularly the national health insurance policy, which they believe fails to adequately address the needs of hypertensive patients. Many noted that the existing health insurance framework does not cover approximately 20% or more of the services required for effective HPT management, compelling patients to purchase essential medications at inflated prices from private pharmacies, thus undermining the fundamental objective of universal health coverage.

##### Subtheme: Policy and Provider Shortages

Participants voiced concerns regarding the limitations of healthcare providers that adversely affect the quality of care in the management of their condition. A common complaint was the inconsistency of healthcare teams at the HPT clinic, which impedes the development of an in-depth understanding of individual patient histories. This inconsistency hampers effective management, as highlighted by one participant:

“*The other barrier to hypertension care is the different doctors we meet at every follow up visit. It is a serious challenge to us as it doesn’t offer that symbiotic patient doctor relationship aim at enhancing medical care for the patient*”[Male HPT patient, 49 years].

Additionally, some participants raised concerns about the competence of healthcare providers and the reliability of blood pressure monitoring equipment, which undermined their confidence in their HPT diagnosis and adherence to treatment.

“*Well, I’m not a hypertension patient, permit me to say that, because sometime when you do a stressful work, your BP is bound to increase. This has been my contest with the results since I was diagnosed hypertensive. I know an error occurred somewhere or their machines are faulty. So, I have never accepted this condition called hypertension that the hospital claimed I have. You see the day I went, I did not relax enough before the examination was done…sometimes their faulty machines declare people with conditions that might not be, and I have protested to the doctor in the consulting room and the nurses are aware of my protest, but they keep given me medication and you think I will take it, no*”[Male HPT patient, 49 years].

##### Subtheme: Lack of Accountability

Participants expressed frustration regarding the perceived lack of transparency and accountability from the government concerning the financial resources generated through health insurance subscriptions. Many questioned the use of their health insurance contributions when they still faced charges for basic outpatient services, laboratory tests, and medications. The participants do not understand why they should subscribe to the NHIS yet will still be charged services they believe should be covered under their insurance subscription, a situation in which they describe the management of the health system resources as opaque and lacking accountability to the citizens.

“*The hospital is partly compounding our plight by continuously introducing charges on services which is affecting some of us. It is unfair and wickedness. So, there is no progress regarding the management of hypertension in this country except that the monies the hospital collect from patients on basic services that we pay for through our subscription to the health insurance have been misapplied to the detriment of the subscriber*”[Female HPT patient, 71 years].

### 3.4. Enablers to Hypertension Management

Four key themes emerged as enablers that enhance the effective management of HPT among patients: acceptance and empowerment, educational empowerment, healthcare access, and policy and provider support.

#### 3.4.1. Theme One: Acceptance and Empowerment

The successful control of HPT is contingent upon both pharmacological and non-pharmacological adherence, which relies on the patient’s capacity for self-management, rooted in acceptance and compliance with recommended medication and lifestyle modifications. Two subthemes were identified: lifestyle modifications and medication adherence.

##### Subtheme: Lifestyle Modifications

Effective management of HPT necessitates adjustments to certain lifestyle behaviours that adversely affect the condition, complementing the use of antihypertensive medications. Many participants reported actively accepting their diagnosis and engaging in recommended practices such as regular physical exercise, reducing or eliminating salt and oily foods, ensuring adequate rest, and avoiding stress. These modifications are vital for enhancing health outcomes.

“*My diet pattern and choices has changed greatly compared to when I was not diagnosed with hypertension. I don’t drink alcohol anymore; I don’t take salty foods and even sugary foods because I have diabetes too. I have also decreased the rate at which I eat fufu. So, I eat fufu “(i.e., a local food from yam)…” on monthly basis and even that I always eat small quantity*”[Female HPT patient, 70 years].

##### Subtheme: Medication Adherence

Participants indicated a strong commitment to adhering to their antihypertensive medication regimens. Many reported not only taking their medications regularly but also adhering to a strict schedule for doses and attending follow-up appointments consistently. Trust in orthodox medicine further supported their adherence behaviours.

“*I follow the medication instructions. I attend reviews anytime they ask me to come and also take drugs every day. I don’t wait for a day to past. For the drugs I take it regularly and timely. I take it twice daily, morning and evening*”[Female HPT patient, 58 years].

#### 3.4.2. Theme Two: Educational Empowerment

This theme highlights the opportunities that empower patients to take control of their health decisions. Two subthemes emerged: health literacy and self-efficacy.

##### Subtheme: Health Literacy

Participants emphasised the role of their educational background in effectively managing HPT. Increased health literacy enabled them to seek information about their condition and make informed health decisions. Many noted a positive correlation between literacy levels and health outcomes.

“*Education is critical as one literacy level helps improve one health literacy level. So, my education has helped me to know my review date and time to attend my reviews, without waiting for someone to remind me. A woman came for review today but was not attended to because her appointment date was not today. If she was educated, she could have seen her actual date for the review that is usually written on the card. So, I think education really help*”[Male HPT patient, 57 years].

##### Subtheme: Self-Efficacy

Participants expressed a strong sense of self-efficacy in managing their HPT, attributing their commitment to lifestyle adjustments and self-management efforts to their educational backgrounds and a desire to maintain health. This commitment is reflected in their willingness to invest both financial and personal resources to achieve control over their condition.

“*I will say my personal abilities/self-efficacy to managing my condition is great. And this is made possible due to my education level, my desire to live so I am committed to every literature and directives the healthcare providers give me*”[Male HPT patient, 40 years].

#### 3.4.3. Theme Three: Healthcare Access and Policy

Access to healthcare and supportive policy frameworks were identified as positive factors in managing HPT. Two subthemes emerged: medication access and policy effectiveness. While some participants noted inefficiencies in the National Health Insurance Scheme (NHIS), others found it beneficial for achieving their healthcare objectives related to HPT management.

##### Subtheme: Medication Access

This subtheme explored participants’ access to antihypertensive medications. Some reported never experiencing shortages of their medications, which positively impacts adherence and control. Instances of running out of medication were often attributed to poor handling rather than supply issues.

“*I haven’t experience drug shortage before coming for next checkup. They usually give us more than the days for next review. Right now, I have old ones here that I want to ask the Doctor if I should continue taking it or discard once I will get new refill*”[Female HPT patient, 74 years].

##### Subtheme: Policy Effectiveness

Participants evaluated the effectiveness of healthcare policies in managing HPT, noting the positive impact of having dedicated clinics for hypertension patients, which reduced waiting times and facilitated easier access to care. Some patients highlighted how policies like the NHIS help subsidise healthcare costs, allowing them to access free antihypertensive medications.

“*I will say to an extent the health policies towards the management of hypertension are somewhat effective, because you can get some drugs and care for free at the public health facilities and some accredited private facilities on the NHIS. I got free drugs the last time and even forgot to find out if I am to continue taking it because I still have leftover of the previous visit. But government is really doing well and is helpful because not everyone can afford the drugs if not given freely*”[Male HPT patient, 54 years].

#### 3.4.4. Theme Four: Provider Support and Relations

The final theme identified comprises two subthemes: family and social support and healthcare provider support. Participants emphasised the crucial role both family support and healthcare provider relationships play in the effective management of hypertension.

##### Subtheme: Family and Social Support

Several participants articulated the critical role of support received from significant others and relatives in managing their hypertension. This support often manifests as financial assistance for purchasing medication and accessing healthcare services, which participants identified as vital enablers in their condition management. Additionally, family members provide essential prompts that aid in medication adherence and the utilisation of follow-up services. One participant noted the following:

“*My children have been supporting me morally and financially. They prompt me on my medication intake and the food I should avoid. It is not only me, but my husband too is also hypertensive, so we have been complementing each other in the management of our condition*”[Female HPT patient, 50 years].

##### Subtheme: Healthcare Provider Support

This subtheme explored the support that healthcare providers offer to patients with HPT, which is crucial for effective condition management. Participants expressed appreciation for the cooperation and assistance received from healthcare providers, highlighting the enhancement of trust and confidence in their care. This support encompasses various services, including health education on nutrition and lifestyle modifications necessary for effective hypertension management. One male participant remarked the following:

“*The health staff at KATH are helping… Some of the health workers are so nice and compassionate…*”[Male HPT patient, 54 years].

## 4. Discussion

This study highlights the multifaceted barriers and enablers affecting HPT management among patients in Ghana. The qualitative nature of this research captures the personal experiences and perceptions of patients, providing a rich understanding of the dynamics involved in HPT management. Our finding identifies five primary barriers: economic burden, environmental and lifestyle influences, knowledge and awareness barrier, medication-related and adherence barrier, and policy and provider-level obstacles. Also, four enablers enhancing HPT control were uncovered: access to healthcare and policy support, acceptance and empowerment, provider support and relationship, and educational empowerment. This suggests the complexity of HPT management, indicating that effective strategies must address both systemic and individual challenges.

The economic burden associated with HPT management emerged as a significant barrier, with participants reporting that the costs of medications, healthcare services, and regular healthcare visits were prohibitive, leading to non-adherence to treatment. This finding is consistent with the existing literature highlighting the financial strain of chronic conditions, particularly in LMICs with limited healthcare resources [60]. Out-of-pocket payments negatively impacted HPT management, as documented in Eriteria [61], India [62], China [63], Tanzania [27], Uganda [21], and Ghana [20,28,64]. Participants attributed the cost burden to the NHIS not covering all aspects of HPT care, underscoring the NHIS’s critical role in mitigating healthcare expenses associated with HPT management [65,66]. The consensus in the findings explains some methodological convergence across the studies compared and the far-reaching impact on HPT management. However, a major contextual factor in Ghana is the partial coverage of HPT care by the NHIS. Many participants in this study expressed concern that NHIS does not fully cover important aspects of HPT management, hence bearing huge financial burdens. This contrasts with some international models where comprehensive public insurance programmes reduce financial barriers to chronic disease care [67]. For instance, in high-income countries with universal health insurance coverage, like Canada and the United Kingdom, antihypertensive medications and regular monitoring are fully covered under health plans, which greatly helps in facilitating adherence to treatment regimens [67]. While financial constraints are usual for most LMICs, Ghana’s economic burden is further increased by the structural inefficiencies within the NHIS. Studies from other African nations such as Nigeria and Kenya bring out similar issues where NHIS provide either partial or inconsistent coverage for chronic conditions, and hence, their effects on the long-term management of diseases remain limited [68,69]. However, Ghana’s NHIS has been commended for being more accessible in terms of coverage compared to most LMICs; the scope for change through policy intervention in terms of coverage extension for HPT-related care will, therefore, be very critical [66]. Policymakers must consider strategies to subsidise medication costs and enhance access to affordable healthcare services, including negotiating lower drug prices or establishing community health initiatives focused on delivering affordable care. Other social interaction factors can be leveraged to attenuate the economic burden of patients on their HPT management through strong social support systems—such as family, friends, and community-based organisations— providing financial help, encouragement, and reminders to patients to take their medication. Besides those mentioned above, the healthcare setting also supports provider relations that are important in overcoming barriers, offering flexible treatment plans, counselling about financials, and improving communication between patients and providers.

The study further highlighted environmental and lifestyle-related limitation as critical barrier to effective HPT management, aligning with previous research that links environmental toxins and unhealthy lifestyle choices to poor HPT control [70,71]. Though our findings are consistent with previous studies, the impact of illegal mining activities, also known as galamsey, on food quality, as reported by participants, is an environmental barrier rarely discussed in the international literature on HPT management. It highlights a localised determinant of health that may require targeted policy interventions different from those seen in other LMICs. For example, small-scale artisanal mining in Ghana has contaminated water and food supplies, worsening health outcomes [72] in a context marked by rising incidences of NCDs and lifestyle changes associated with urbanisation [73]. Limited access to exercise facilities further exacerbates these challenges, especially in urban settings where sedentarism is on the rise. Alongside community-level initiatives that improve access to healthy foods and safe exercise environments, public health campaigns promoting healthy eating and physical activity could mitigate these issues. We recommend that the Ghanaian government address the detrimental effects of galamsey activities and implement effective land reclamation measures. Conversely, the study identified provider support and positive familial relationships as enablers, which can help mitigate the environmental and sociocultural pressures affecting lifestyle-related limitations. Our findings confirm the literature that suggests family and social support can serve as effective measures against lifestyle and other environmental challenges in HPT management [18,27,61]. Family support is vital in facilitating access to balanced, low-sodium diets and addressing dietary limitations, especially in areas with limited healthy food options. Family members also promote consistent physical activity through encouragement and accountability, thereby making exercise more sustainable even in resource-constrained settings. Furthermore, emotional and practical support from family members reduces stress—an essential factor in HPT control—by alleviating daily burdens such as transportation to medical appointments. The combined involvement of healthcare providers and family members enhances medication adherence and improves health literacy and consistency in medication use. This social support network facilitates adaptive responses to environmental limitations, emphasising the importance of family-centred approaches for effective HPT management.

The study identified a significant lack of knowledge and awareness regarding HPT and its management among participants. Similar findings have been observed in studies conducted in Namibia [19], Nepal [18], and China [63], where participants expressed uncertainty and poor knowledge regarding HPT management practices. This lack of awareness is concerning as it can inhibit patients from taking proactive measures for early diagnosis and management of their conditions. For example, Bhattarai et al. [18] found that inadequate knowledge of HPT among patients often led to medication non-adherence. Similarly, research in Malaysia indicated that knowledge deficits regarding HPT resulted in poor blood pressure monitoring opportunities [74]. These findings highlight the pervasive misconceptions and lack of awareness surrounding HPT and its management across cultures. We infer that the absence of formal education may exacerbate barriers to HPT control in Ghana, as many participants reported minimal or no formal education. However, our findings also suggest a pathway to address these barriers. Participants indicated that educational empowerment through increased health literacy and self-efficacy could enable better management of their HPT. Many participants noted that their formal education had equipped them with essential health literacy skills and boosted their self-efficacy, thereby enhancing their ability to control their HPT conditions. This aligns with previous research conducted in LMICs [61,75,76,77] and Asia [16,18,25], which emphasised the importance of knowledge empowerment in facilitating effective self-management practices for HPT. We advocate for intensified health education campaigns aimed at empowering patients to take an active role in managing their hypertension. Implementing educational programmes within healthcare settings and community platforms can enhance patient understanding, ultimately leading to improved adherence and better health outcomes [78,79]. It may also be beneficial to tailor educational materials to accommodate cultural and linguistic contexts, ensuring they are accessible and relevant to diverse populations [78,79]. Additionally, adopting a personalised health education strategy that involves a multidisciplinary team—comprising pharmacists, nurses, doctors, and community health practitioners—can enhance patients’ health literacy and empowerment, yielding positive outcomes in hypertension management [80,81].

Our findings indicate that medication-related barriers significantly challenge patients’ self-management and their ability to achieve hypertension control goals. Additional sub-themes, including non-adherence and medication shortages, underscore patients’ concerns regarding the challenges of antihypertensive medications’ impact on HPT management. This finding points to significant factors hindering effective HPT control, including medication side effects (e.g., frequent urination, reduced libido in men, sleep disturbances, and headaches), forgetfulness, and shortages of medications at government hospital pharmacies affecting patients’ refills. This medication-related adherence barrier has been well-documented in the literature as a significant obstacle to the pharmacological management of HPT [16,19,20,28,70]. For example, medication side effects have been reported as obstacles to adherence across studies conducted in Iran [70], Nepal [16], Namibia [19], and Ghana [20,28]. Furthermore, medication shortages [19,20,28] and forgetfulness [19,70] further complicate adherence to medication regimens. These findings highlight the need for healthcare providers to engage patients not only in prescribing medications but also in discussing their treatment plans, including potential side effects and the importance of adherence. Strategies such as simplifying treatment regimens, developing reminders for medication dosing, and conducting regular follow-ups can significantly improve adherence. Furthermore, involving patients in shared decision-making regarding their treatment can enhance adherence by fostering a sense of ownership over their health management. Regarding medication shortages, we recommend that the Ministry of Health and the authorities at KATH explore better supply chain solutions to ensure consistent availability of essential antihypertensive medications at government health facilities.

Participants in the present study identified opportunities to address medication-related adherence barriers by highlighting that effective medication access and policy can significantly enhance adherence to treatment regimens and overall HPT management. Patients who consistently reported adequate medication refills noted a corresponding adherence to their medication schedules, suggesting that access to medicines is crucial for facilitating HPT control. This observation is consistent with findings from previous studies [16,18,19], which indicate that improved availability of services and free healthcare provisions enhance the utilisation of services among HPT patients, thereby yielding positive management outcomes. Participants noted that the NHIS has been instrumental in enabling access to subsidised healthcare services and free antihypertensive medications, which contribute to adherence and effective HPT management. To achieve the goal of a 25% reduction in HPT burden by 2025, it is essential to improve the NHIS by expanding coverage to encompass more health services and allowing medical practitioners greater flexibility in prescribing a wider range of medications [10].

Additionally, the study found that patient acceptance and empowerment are critical enablers that can help address medication-related adherence issues. Those who accepted their HPT diagnosis expressed greater self-confidence in adopting lifestyle modifications and committing to medication adherence. Many individuals reported that understanding their diagnosis motivated them to make healthier lifestyle choices and seek necessary medical care. Empowerment initiatives, such as self-management training programmes, can further facilitate this process by equipping patients with the skills and knowledge necessary for effective HPT control.

The study also identified policy and provider-level barriers reflected in participants’ experiences within the Ghanaian healthcare system. Patients highlighted gaps in policy and provider accountability regarding the resources mobilised through the NHIS. Some participants expressed concerns that the frequent changes in medical personnel during follow-up visits adversely affect the effective management of their conditions. Moreover, patients reported instances of inadequate training among some healthcare staff, particularly nurses, in accurately capturing vital signs, compounded by issues with faulty or poor-quality blood pressure monitoring devices, which engendered mistrust in their diagnostic results. Although the literature contrasting these findings are limited, HPT patients in Nepal have reported similar experiences, citing unsupportive provider behaviours that undermine trust in the care received [16,18]. Other research within the Ghanaian context corroborates our findings [28,29]. For example, lapses in the NHIS coverage for certain HPT-related care and medications, as well as delays in reimbursing healthcare providers for services rendered, have been identified as significant barriers to achieving HPT control in Ghana. This observation has been echoed in the analysis conducted by the Institute of Statistical, Social and Economic Research (ISSER) [66], which outlined deficiencies within the NHIS in meeting patients’ healthcare financing needs. Furthermore, a scoping review assessing Ghana’s healthcare policy regarding HPT found it to be suboptimal and counterproductive to achieving the target of 25% HPT control by 2025 [82]. Evidence-based reforms are hence necessary to deal with such challenges. First, extending NHIS coverage to include comprehensive HPT services, including routine screening and essential medications, will reduce out-of-pocket expenses. This has been seen to work in other LMICs, such as Rwanda, where universal health coverage improved the management of chronic diseases [83]. The second reduction in delay will also occur because reimbursement procedures through digitalised payment systems will facilitate health facilities in general—a feature piloted in Kenya [84] and ensured by community-based interventions using trained community health workers (CHWs) for HPT screening and management, with established gaps in NCDs across the sub-Saharan Africa (SSA) countries [85,86]. Finally, the integration of HPT management into primary healthcare and the adoption of multisectoral approaches to address the social determinants of health, as witnessed in Thailand’s NCD programmes [87], might further bolster Ghana’s health policy framework.

Participants also expressed concerns about the reliability of the blood pressure equipment used for monitoring and decision-making at hospitals, a sentiment echoed by Nyaaba et al. [29]. The similarities in these findings underscore broader geo-economic challenges and inequities within the healthcare system and their consequent impact on HPT management.

We recommend that the Ghanaian government shift its focus from merely meeting spending targets to evaluating how resources can be effectively allocated to improve population health and achieve universal health coverage. This involves assessing whether funds are being utilised efficiently to advance community health and identifying ways to improve resource allocation and reduce waste [88]. These issues underscore the need for healthcare systems to foster patient-centred environments that enhance communication and trust between healthcare providers and patients. Training healthcare providers in effective communication skills and patient engagement strategies can cultivate a supportive atmosphere that improves patient satisfaction and adherence.

Additionally, medication access and policy effectiveness emerged as vital enablers in the present study. Participants indicated that existing healthcare policies are effective in managing HPT, crediting the NHIS for providing free medications and healthcare services, which they viewed as essential for successful HPT management. We encourage the Ghanaian government and healthcare providers to continue expanding such support to encompass all HPT-related services, including branded antihypertensive medications, which some patients reported as having fewer side effects and thus improving adherence.

### 4.1. Strengths and Limitations

This study provides a nuanced understanding of the barriers and enablers in HPT management in Ghana by capturing patients’ experiences via a qualitative approach. Key strengths include the identification of specific, intersecting barriers such as economic burdens, lifestyle and environmental factors, limited awareness, medication adherence issues, and healthcare policy constraints. By elucidating these challenges alongside enablers such as healthcare access, social support, educational empowerment, and policy interventions, the study highlights the need for a comprehensive, patient-centred approach. The findings enhance our understanding of chronic disease management in low- and-middle-income countries like Ghana and offer insights that could inform targeted interventions and policies in similar contexts.

However, this study has limitations including the potential for social desirability bias. Additionally, the sample may not fully represent the diversity of the Ghanaian HPT patient population, limiting the transferability of the findings. Moreover, the focus on patient perspectives may omit critical provider-side challenges in HPT management. Future research could address these limitations by employing mixed method approaches that include healthcare providers’ perspectives, thereby providing a more comprehensive view of the systemic factors affecting HPT management in Ghana.

### 4.2. Implications for Policy and Practice

This research highlights important considerations for policymakers and healthcare providers in Ghana concerning the management of HPT. It outlines the necessity for policymakers to enhance the NHIS to alleviate the economic burden on patients by providing comprehensive coverage for HPT-related medications and medical services. Additionally, robust public health campaigns are essential to foster health literacy and awareness about HPT, encouraging patients to take proactive roles in their self-management efforts. Furthermore, our findings advocate for the integration of family, social, and provider support into clinical practices, recognising their significant role in promoting adherence and facilitating lifestyle modifications for effective HPT management. The study underscores the importance of community initiatives aimed at increasing access to safe spaces for physical activity and promoting healthy food options to address lifestyle and environmental limitations. Moreover, training healthcare providers in effective communication and engagement strategies can enhance provider–patient relationships, ultimately leading to improved patient satisfaction and adherence to treatment. Lastly, ongoing evaluation of healthcare services is critical to ensuring that HPT patients achieve their self-management goals and realise better health outcomes.

## 5. Conclusions

This research highlights the various obstacles and facilitators influencing the management of HPT among patients in Ghana. Our study explores the complex interplay between economic conditions, healthcare policies, environmental factors, and patient knowledge levels, all of which significantly affect treatment adherence and disease control outcomes. We specifically highlighted the detrimental impact of financial constraints, particularly high out-of-pocket costs and limitations within the NHIS, on patients’ health aspirations. Furthermore, lifestyle factors and a lack of information exacerbate the challenges faced by individuals striving to manage their HPT effectively.

Conversely, the research identifies several critical facilitators, including access to healthcare, empowerment through education, support from healthcare providers, and social connections, which can substantially enhance patient outcomes when employed effectively. These findings underscore the necessity for a patient-centred approach that addresses both systemic issues and individual barriers in the management of hypertension. 

Consequently, it is essential for policymakers to reform existing healthcare policies, strengthen public health initiatives, and foster collaborative relationships between patients and healthcare providers. Such measures would create a comprehensive framework for effective management of HPT. Ultimately, the study’s findings advocate for holistic strategies that empower individuals to improve their health literacy and ensure equitable access to essential resources, thereby advancing HPT control and enhancing public health across Ghana. Future research should consider investigating the effects of multi-level interventions on improving HPT management outcomes within the Ghanaian context.

## Figures and Tables

**Figure 1 healthcare-13-00479-f001:**
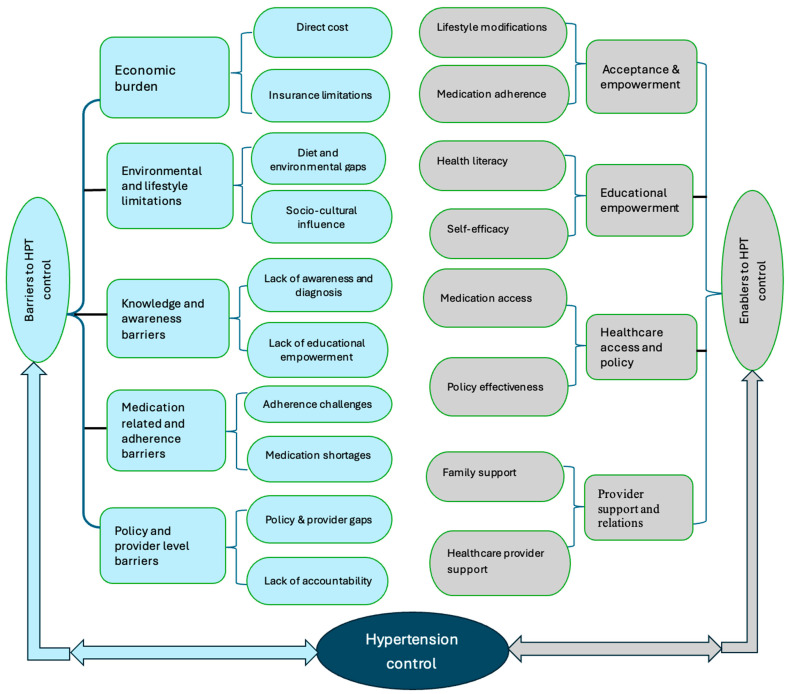
Diagrammatic representation of themes and subthemes.

## Data Availability

The datasets used and/or analysed during the current study are available from the corresponding authors upon reasonable request.

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
