# Peer review of "A Qualitative Study on the Barriers and Enablers to Effective Hypertension Management in Ghana"

_healthcare, 2025, doi:10.3390/healthcare13050479_

Round 1
Reviewer 1 Report
Comments and Suggestions for Authors
Peer review report for the manuscript entitled A Qualitative Study on the Barriers and Enablers to Effective Hypertension Management in Ghana.
This manuscript describes a qualitative study assessing hypertension nurses' perspectives on barriers and enablers to effectively managing the condition. The manuscript was well-written and organized, even though occasionally, word choices were ambiguous. I offer some comments that could help clarify certain aspects and enhance the overall quality of the manuscript.
This manuscript presents a qualitative study examining patients' perspectives on the barriers and facilitators affecting their ability to manage the condition effectively. Overall, the manuscript is well-written and organized; however, there are moments where word choices lack clarity. Also, I have some questions about the methods that need clarification.
I have provided some comments that could help clarify certain aspects and improve the overall quality of the manuscript.
1. Line 14 in the Abstract (and elsewhere) – Suggest removing “lived” (experience).
2. Line 99 – Suggest rephrasing “to manage hypertension effectively.”
3. Line 103 (and elsewhere) – Suggest removing “chronic” (hypertension).
4. Subsection 1.1. Conceptual Framework
- The authors mention repeatedly that this manuscript is part of a more extensive study. Additionally, they note that the exclusion of several participants is based on the findings from the quantitative portion of this larger study. I recommend that the authors provide a brief introduction to the larger study, including its objectives, sponsors, methods of measurement, and relevant results from other parts, if applicable.
- Lines 116-117 – The authors state, “It (the Chronic Care Model) underscores the critical role of ‘self-management’ by patients.” I would appreciate it if the authors could elaborate on this statement. The Chronic Care Model consists of six fundamental components, with “self-management support” being one of the most important. The authors may have focused their use of the Chronic Care Model on this particular component while other parts of the “larger study” address the remaining components.
- Lines 121-124 seem to contradict the last sentence of the previous paragraph, suggesting that a broader investigation into hypertension management is necessary rather than focusing solely on the patients’ self-management efforts.
5. Subsections 2.2, "Study Setting, Design, and Population," and 2.3, "Target Population and Sample," could benefit from clarification. Subsection 2.2 is somewhat ambiguous, while subsection 2.3 repeats content found in 2.2. I suggest the authors reorganize these sections to clearly describe the study settings, study design, and participant recruitment.
6. What are the inclusion and exclusion criteria for participants’ recruitment? Is the target population the quantitative study’s sample? (I reiterate the importance of briefly introducing the larger study and its other parts.)
7. Are the participants adults (18+ years of age)? Are there any adolescent children below the age of 18? The authors must clarify the age range of the participants. Parents' consent must also be documented in the manuscript if children are included.
8. Lines 164-165 need to elaborate on the principle of data saturation and how sample size is calculated.
9. Lines 167-168, why were those participants excluded? (inclusion and exclusion criteria?)
10. Lines 168-169, what is the instrument? I suggest the authors provide the instruments as in appendices.
11. Data collection through “semi-structured interview”?
12. The Results Section:
- In the subtheme of Theme 2, Diet and environmental factors, the authors present the participants’ perceptions of how the environment affects food but do not connect the environment with the participants' diet.
- Subtheme “Lack of awareness and diagnosis” presents an undiagnosed issue due to a lack of transparent communication between patients and providers.
The subtheme “lack of educational empowerment” is likely a low health literacy issue due to education.
- Subthemes Policy & lack of accountability seem to repeat health insurance issues.
- I suggest replacing “provider gaps” with “provider shortage.”
I congratulate the authors on the work that was well done.
Author Response
This manuscript describes a qualitative study assessing hypertension nurses' perspectives on barriers and enablers to effectively managing the condition. The manuscript was well-written and organized, even though occasionally, word choices were ambiguous. I offer some comments that could help clarify certain aspects and enhance the overall quality of the manuscript.
This manuscript presents a qualitative study examining patients' perspectives on the barriers and facilitators affecting their ability to manage the condition effectively. Overall, the manuscript is well-written and organized; however, there are moments where word choices lack clarity. Also, I have some questions about the methods that need clarification.
Response: We are grateful for your comforting comments.
I have provided some comments that could help clarify certain aspects and improve the overall quality of the manuscript.
- Line 14 in the Abstract (and elsewhere) – Suggest removing “lived” (experience).
Response: We are grateful for your comment. We have removed “lived” in the sentence and all other places it appeared. See page 1 line 14 for details.
- Line 99 – Suggest rephrasing “to manage hypertension effectively.”
Response: Thank you for your suggestion. We have amended the sentence to incorporate your suggestion to now read “to manage hypertension effectively”. See page 2 line 99 for details.
- Line 103 (and elsewhere) – Suggest removing “chronic” (hypertension).
Response: Thank you for your suggestion. We have removed the word “chronic” from the sentence. See page 2 line 102 for details.
- Subsection 1.1. Conceptual Framework
The authors mention repeatedly that this manuscript is part of a more extensive study. Additionally, they note that the exclusion of several participants is based on the findings from the quantitative portion of this larger study. I recommend that the authors provide a brief introduction to the larger study, including its objectives, sponsors, methods of measurement, and relevant results from other parts, if applicable.
Response: Thank you for your suggestion. We provided additional details to the larger study including its objectives. This study was conducted at the Komfo Anokye Teaching Hospital (KATH) in the Ashanti Region (A/R) of Ghana. KATH is one of the major referral hospitals serving a large portion of the population in Ghana. The A/R was selected due to its high prevalence of HPT with epidemiological data showing 15.3% of women and 18.1% of men between 15–49 years with HPT, the highest in Ghana [38]. KATH is also famous for its pool of skilled healthcare providers on HPT patients’ management [39] and provide an opportunity to access HPT patients with varying degrees of severity, allowing the researchers to have a comprehensive assessment of the enablers and barriers affecting HPT management in Ghana.
This study is part of a larger project that evaluated Ghana’s progress towards HPT control in the A/R. The study seeks to address the following objectives: To assess the extent of Ghana’s adherence to the PASCAR’s 10-point action plan towards HPT control; To assess patient barriers affecting HPT control in Ghana; To evaluate government and health system barriers on HPT control in Ghana; and to explore stakeholders’ perspectives on barriers and enablers to HPT control. However, this study used descriptive qualitative design [40] and an inductive thematic analysis framework [41] to explore HPT patients perceived enablers and barriers to HPT management. This design allows for a detailed examination of the participants’ experiences, perceptions, and challenges in managing their condition and thus capture their contextual insights that quantitative methods may overlook. The study followed the Consolidated Criteria for Reporting Qualitative Studies (COREQ) guidelines [42]. More details are provided in the supplementary material. See page 151-171 for details.
Lines 116-117 – The authors state, “It (the Chronic Care Model) underscores the critical role of ‘self-management’ by patients.” I would appreciate it if the authors could elaborate on this statement. The Chronic Care Model consists of six fundamental components, with “self-management support” being one of the most important. The authors may have focused their use of the Chronic Care Model on this particular component while other parts of the “larger study” address the remaining components.
Response: We are grateful to you for your insightful comment. We have revised the model description to incorporate your suggestion on the “self-management” and also focused it on the patient level construct. The revised section now reads, “The major components of the model are six independent constructs [14]. Four (4) of the constructs; “health system organization”, “delivery system support”, “decision support”, and “clinical information support” are concerned with the practices of healthcare providers, whereas the other two; “self-management” and “community resources and policies” concerns the patients [31]. For this present study, the “self-management” (i.e. refers to the actions and strategies that patients adopt to control their blood pressure, reduce complications, and improve their overall well-being) by patients [14,32], which encompasses both pharmacological and non-pharmacological strategies essential for effective control of HPT was utilized [33]. The “self-management support” in the model has been adapted for this study to imply HPT patients’ disease management practices which has been found to be limited by barriers such as lack of medication adherence, poor knowledge on antihypertensive issues and lifestyle modification challenges [10]. The present study therefore evaluated these patient level barriers and their potential to hinder Ghana’s progress towards HPT control and sort to unpack enablers to mitigate barriers affecting HPT patients’ “self-management” practices with effectual improved health outcomes. Consequently, this study explored the experiences of HPT patients, focusing on the enablers and barriers that influence their management efforts. The CCM is particularly appropriate for this investigation, as it facilitates the assessment of the self-management practices of HPT patients to unpack the barriers and enablers impacting HPT management. Moreover, this model has been widely validated in similar studies internationally [30,34,35] and applied within the Ghanaian context [36]. See page 3 lines 116-137 for details.
Lines 121-124 seem to contradict the last sentence of the previous paragraph, suggesting that a broader investigation into hypertension management is necessary rather than focusing solely on the patients’ self-management efforts.
Response: Thank you for your comment. We have revised the conceptual framework to incorporate your suggestion. See page 3 lines 116-137 for details.
- Subsections 2.2, "Study Setting, Design, and Population," and 2.3, "Target Population and Sample," could benefit from clarification. Subsection 2.2 is somewhat ambiguous, while subsection 2.3 repeats content found in 2.2. I suggest the authors reorganize these sections to clearly describe the study settings, study design, and participant recruitment.
Response: Thank you for your suggestions. We have reorganised the various sections to now read, as below.
Study setting and design
This study was conducted at the Komfo Anokye Teaching Hospital (KATH) in the Ashanti Region (A/R) of Ghana. KATH is one of the major referral hospitals serving a large portion of the population in Ghana. The A/R was selected due to its high prevalence of HPT with epidemiological data showing 15.3% of women and 18.1% of men between 15–49 years with HPT, the highest in Ghana [38]. KATH is also famous for its pool of skilled healthcare providers on HPT patients’ management [39] and provide an opportunity to access HPT patients with varying degrees of severity, allowing the researchers to have a comprehensive assessment of the enablers and barriers affecting HPT management in Ghana.
This study is part of a larger project that evaluated Ghana’s progress towards HPT control in the A/R. The study seeks to address the following objectives: To assess the extent of Ghana’s adherence to the PASCAR’s 10-point action plan towards HPT control; To assess patient barriers affecting HPT control in Ghana; To evaluate government and health system barriers on HPT control in Ghana; and to explore stakeholders’ perspectives on barriers and enablers to HPT control. However, this study used descriptive qualitative de-sign [40] and an inductive thematic analysis framework [41] to explore HPT patients perceived enablers and barriers to HPT management. This design allows for a detailed ex-amination of the participants’ experiences, perceptions, and challenges in managing their condition and thus capture their contextual insights that quantitative methods may over-look. The study followed the Consolidated Criteria for Reporting Qualitative Studies (COREQ) guidelines [42]. More details are provided in the supplementary material. Find details in pages 4 lines 140-160.
Target population and sample
The participant cohort comprised a diverse group of adults diagnosed with HPT, varying in age, gender, socioeconomic status, and educational background. This diversity ensured a comprehensive understanding of the factors influencing HPT management. A total of twenty (20) HPT patients were recruited, selected conveniently during a pre-interview survey. This sample size was arrived at guided by the principle of data saturation where additional interviews/data collection efforts no longer reveal new information, suggesting that additional sampling is not needed [43]. The adequacy of this sample size aligns with existing literature in qualitative studies in health [44,45] and specifically, chronic disease management research [45] that demonstrates that data saturation could be achieved with 12 interviews in a homogeneous population [44] and between 10–20 participants when methodological rigour is maintained. The determination of sample size was guided by the principle of data saturation [43]. This ensured our study was complete in experiences and perspectives, and unconstrained by an arbitrary sample size thereby lending more credibility and rigor to the findings.
Participants included HPT patients registered with the KATH HPT clinic who had attended follow-up appointments for at least one year and had previously participated in the earlier quantitative data collection process as part of the broader project. Additionally, HPT patients who were on admission but had taken part in the earlier quantitative data collection process, alert, and provided consent were included in the study. Including only those who had engaged in the research earlier ensured that the participants indeed had relevant medical histories and experiences with HPT management at KATH, which in turn made their experiences more reliably comparable. HPT patients who were not registered members of the KATH HPT clinic, who did not participate in the earlier quantitative study phase and did not consent were excluded. This was done to maintain consistency and continuity in data collection. Those who did not consent were excluded in order not to violate ethical standards and respect for participants' autonomy. See page 4 lines 173-199.
- What are the inclusion and exclusion criteria for participants’ recruitment? Is the target population the quantitative study’s sample? (I reiterate the importance of briefly introducing the larger study and its other parts.)
Response: We are glade for your insightful observations and have addressed the issues on the inclusion and exclusion criteria as follows, “The participant cohort comprised a diverse group of adults diagnosed with HPT, varying in age, gender, socioeconomic status, and educational background. This diversity ensured a comprehensive understanding of the factors influencing HPT management. A total of twenty (20) HPT patients were recruited, selected conveniently during a pre-interview survey. This sample size was arrived at guided by the principle of data saturation where additional interviews/data collection efforts no longer reveal new information, suggesting that additional sampling is not needed [43]. The adequacy of this sample size aligns with existing literature in qualitative studies in health [44,45] and specifically, chronic disease management research [45] that demonstrates that data saturation could be achieved with 12 interviews in a homogeneous population [44] and between 10–20 participants when methodological rigour is maintained. The determination of sample size was guided by the principle of data saturation [43]. This ensured our study was complete in experiences and perspectives, and unconstrained by an arbitrary sample size thereby lending more credibility and rigor to the findings.
Participants included HPT patients registered with the KATH HPT clinic who had attended follow-up appointments for at least one year and had previously participated in the earlier quantitative data collection process as part of the broader project. Additionally, HPT patients who were on admission but had taken part in the earlier quantitative data collection process, alert, and provided consent were included in the study. Including only those who had engaged in the research earlier ensured that the participants indeed had relevant medical histories and experiences with HPT management at KATH, which in turn made their experiences more reliably comparable. HPT patients who were not registered members of the KATH HPT clinic, who did not participate in the earlier quantitative study phase and did not consent were excluded. This was done to maintain consistency and continuity in data collection. Those who did not consent were excluded in order not to violate ethical standards and respect for participants' autonomy.” See page 4 lines 187-198 for details. For your suggestion on an introduction to the broader project, please see page 4 lines 160-170 for details.
- Are the participants adults (18+ years of age)? Are there any adolescent children below the age of 18? The authors must clarify the age range of the participants. Parents' consent must also be documented in the manuscript if children are included.
Response: We are grateful to you for your comment. There was no adolescent, and the age ranged of the participants is indicated in the manuscript as, “Participants’ ages ranged from 48 to 89 years” under the results section in page 6 lines 297-298.
- Lines 164-165 need to elaborate on the principle of data saturation and how sample size is calculated.
Response: Thank you for your comment. We have elaborated on your comment as follows: “The participant cohort comprised a diverse group of adults diagnosed with HPT, varying in age, gender, socioeconomic status, and educational background. This diversity ensured a comprehensive understanding of the factors influencing HPT management. A total of twenty (20) HPT patients were recruited, selected conveniently during a pre-interview survey. This sample size was arrived at guided by the principle of data saturation where additional interviews/data collection efforts no longer reveal new information, suggesting that additional sampling is not needed [43]. The adequacy of this sample size aligns with existing literature in qualitative studies in health [44,45] and specifically, chronic disease management research [45] that demonstrates that data saturation could be achieved with 12 interviews in a homogeneous population [44] and between 10–20 participants when methodological rigour is maintained.” See page 4 lines 173-183 for details.
- Lines 167-168, why were those participants excluded? (inclusion and exclusion criteria?)
Response: Thank you so much for your inquest. Reasons for the inclusion and exclusion criteria have been included as “Participants included HPT patients registered with the KATH HPT clinic who had attended follow-up appointments for at least one year and had previously participated in the earlier quantitative data collection process as part of the broader project. Additionally, HPT patients who were on admission but had taken part in the earlier quantitative data collection process, alert, and provided consent were included in the study. Including only those who had engaged in the research earlier ensured that the participants indeed had relevant medical histories and experiences with HPT management at KATH, which in turn made their experiences more reliably comparable. HPT patients who were not registered members of the KATH HPT clinic, who did not participate in the earlier quantitative study phase and did not consent were excluded. This was done to maintain consistency and continuity in data collection. Those who did not consent were excluded in order not to violate ethical standards and respect for participants' autonomy.” See page 4 lines 194-198 for the reasons for exclusion.
- Lines 168-169, what is the instrument? I suggest the authors provide the instruments as in appendices.
Response: Thank you. We have included the interview guide as an appendix and referenced it appropriately “The interview guide has been attached as an appendix (Appendix 1).” Page 5 lines 205-206 for details.
- Data collection through “semi-structured interview”?
Response: Thank you for your observation. We have rectified the sentence. See page 5 lines 202-209 for details.
- The Results Section:
In the subtheme of Theme 2, Diet and environmental factors, the authors present the participants’ perceptions of how the environment affects food but do not connect the environment with the participants' diet.
Response: We are grateful to you for your observation. We have amended by connecting the negative impact of the environmental factor as indicated by the participants to their dietary and hypertension condition and management. The revised version now reads, “Many participants reported that their dietary habits negatively impacted their ability to manage HPT. Some attributed their challenges in achieving successful HPT management to their diet, while others blamed illegal mining activities (locally referred to as galamsey), which introduce harmful chemical residues into the soil and subsequently into food crops. This perception underlines the greater implications of environmental degradation for public health. It is not only the compromised quality of diets due to toxic substances in food sources but also the increased risk of hypertension-related complications. Thus, effective management of hypertension requires not only personal adjustment in the diet but also systemic interventions in the form of regulating environmental hazards that impact on food safety.” See page 8 lines 360-365.
Subtheme “Lack of awareness and diagnosis” presents an undiagnosed issue due to a lack of transparent communication between patients and providers.
Response: We appreciate your observation. We have amended the subtheme to now read, “Patients reported a significant lack of knowledge and awareness of the precursors to HPT, which has contributed to detrimental health behaviours and delayed diagnosis. Many expressed frustrations over the late diagnosis of their condition, which adversely affected their management efforts. This reflects the bigger problem: lack of proper aware-ness and consequently late diagnosis—that has impeded the effective control of HPT and hence its complications among the patients. So, once people lack knowledge about the early signs and symptoms relative to HPT, they may resort to unorthodox modes of man-aging their ill health leading to late diagnosis. This is evident in the submission of one of the participants as.” See page 9, lines-412-417 for details.
The subtheme “lack of educational empowerment” is likely a low health literacy issue due to education.
Response: We acknowledged your candid suggestion and have rectified the section to now read, “Participants discussed how a lack of formal education or literacy hampers their ability to monitor their health and engage effectively with healthcare services. The participant's account revealed the influence of low formal education on effective HPT management. Illiteracy prevents patients from reading about their disease, checking the medicines prescribed for them, and even remembering their next appointments. The participant con-fessed that education is important in helping one to identify possible mistakes in their care and to make appropriate decisions concerning their health. Without this knowledge, many patients, like him, have to depend on others for help, increasing their susceptibility to medication errors and mismanagement. This shows the need for patient education programmes so as to improve self-management and increase health outcomes. This was highlighted by one of the HPT patients as described hereto.” Page 9 lines 425-434 for details.
Subthemes Policy & lack of accountability seem to repeat health insurance issues.
Response: Thank you for your observation. It is true that some aspects of health insurance issues were discussed. However, this section was about how the health insurance mobilisations were not properly accounted for towards patients’ healthcare services provision as patients laments opaqueness and lack of accountability with the insurance funds. See page 10 lines 495-501 for details.
I suggest replacing “provider gaps” with “provider shortage.”
Response: We appreciate your suggestion, and we have incorporated it appropriately. See page 10 line 502 for details.
I congratulate the authors on the work that was well done.
Response: We express our gratitude to you for your high recommendations.

Reviewer 2 Report
Comments and Suggestions for Authors
This manuscript, titled “a qualitative study on the barriers and enablers to effective hypertension management in Ghana” explores the barriers and enablers to effective hypertension (HPT) management among patients in Ghana. This paper utilized a qualitative research design guided by the Chronic Care Model. Overall, the paper offers valuable insights into the economic, socio-cultural, and policy factors influencing HPT management in a low-resource setting. The thematic presentation of both barriers (e.g., medication costs, weak health insurance coverage, environmental and lifestyle constraints, knowledge gaps) and enablers (e.g., family support, acceptance and empowerment, educational opportunities, healthcare access) is well-structured and coherent. The study underscores the importance of patient-centered interventions, improved health insurance policy, and tailored health education campaigns to achieve better HPT outcomes. While the paper is timely and relevant to global discussions on chronic disease control, there are specific areas where revisions can further strengthen its contribution.
1. The paper mentions recruiting 20 participants to achieve data saturation; however, further elaboration on the sampling technique (e.g., convenience, purposive, or snowball) and a justification of how saturation was determined would strengthen the methodological transparency. Clarifying inclusion and exclusion criteria will also help readers assess transferability.
2. The in-depth interviews are well-described, but additional details about how the interview guide was developed (e.g., based on the Chronic Care Model or prior literature) would be beneficial. An appendix with key interview questions or prompts could enhance the manuscript’s completeness.
3. In qualitative studies, demonstrating credibility, dependability, confirmability, and transferability is crucial. Incorporating information on how you managed reflexivity (e.g., researcher positionality) or how you triangulated data (if at all) will further validate your findings. Providing more on inter-coder reliability or member-checking procedures would also increase confidence in the results.
4. While the paper outlines an inductive thematic approach, a more detailed step-by-step description of the coding and theme development process would be helpful. For instance, clarify how the research team resolved disagreements about coding or theme labels, and describe how themes were refined to best represent the data.
5. The manuscript mentions multiple factors impacting hypertension management. It might be instructive to discuss how these factors intersect or reinforce each other. A brief section on the interactions between economic burden and knowledge deficits, or how social support can mitigate economic barriers, would enrich the discussion.
6. The findings that highlight limitations in Ghana’s National Health Insurance Scheme (NHIS) and broader health policies are important. Strengthening this section with concrete, evidence-based recommendations—such as specific policy reforms or scalable community interventions—would add real-world applicability to your results.
7. The references to other African or LMIC studies are informative, but expanding on how your findings align or diverge from international literature would demonstrate the study’s broader relevance. Highlighting unique Ghanaian contextual factors vs. more universal barriers and enablers could solidify the study’s global health significance.
Author Response
Reviewer 2
This manuscript, titled “a qualitative study on the barriers and enablers to effective hypertension management in Ghana” explores the barriers and enablers to effective hypertension (HPT) management among patients in Ghana. This paper utilized a qualitative research design guided by the Chronic Care Model. Overall, the paper offers valuable insights into the economic, socio-cultural, and policy factors influencing HPT management in a low-resource setting. The thematic presentation of both barriers (e.g., medication costs, weak health insurance coverage, environmental and lifestyle constraints, knowledge gaps) and enablers (e.g., family support, acceptance and empowerment, educational opportunities, healthcare access) is well-structured and coherent. The study underscores the importance of patient-centered interventions, improved health insurance policy, and tailored health education campaigns to achieve better HPT outcomes. While the paper is timely and relevant to global discussions on chronic disease control, there are specific areas where revisions can further strengthen its contribution.
Response: We expressed our profound gratitude to you for your quality time in reviewing our manuscript and for your constructive and insightful comments and suggestions.
- The paper mentions recruiting 20 participants to achieve data saturation; however, further elaboration on the sampling technique (e.g., convenience, purposive, or snowball) and a justification of how saturation was determined would strengthen the methodological transparency. Clarifying inclusion and exclusion criteria will also help readers assess transferability.
Response: We are glade for your guidance and have clarified by including justification for the sample size and how saturation was determined as well as the inclusion and exclusion criteria of the study. The new rendition to the concern sections now reads, “The participant cohort comprised a diverse group of adults diagnosed with HPT, varying in age, gender, socioeconomic status, and educational background. This diversity ensured a comprehensive understanding of the factors influencing HPT management. A total of twenty (20) HPT patients were recruited, selected conveniently during a pre-interview survey. This sample size was arrived at guided by the principle of data saturation where additional interviews/data collection efforts no longer reveal new information, suggesting that additional sampling is not needed [43]. The adequacy of this sample size aligns with existing literature in qualitative studies in health [44,45] and specifically, chronic disease management research [45] that demonstrates that data saturation could be achieved with 12 interviews in a homogeneous population [44] and between 10–20 participants when methodological rigour is maintained. The determination of sample size was guided by the principle of data saturation [43]. This ensured our study was complete in experiences and perspectives, and unconstrained by an arbitrary sample size thereby lending more credibility and rigor to the findings.
Participants included HPT patients registered with the KATH HPT clinic who had attended follow-up appointments for at least one year and had previously participated in the earlier quantitative data collection process as part of the broader project. Additionally, HPT patients who were on admission but had taken part in the earlier quantitative data collection process, alert, and provided consent were included in the study. Including only those who had engaged in the research earlier ensured that the participants indeed had relevant medical histories and experiences with HPT management at KATH, which in turn made their experiences more reliably comparable. HPT patients who were not registered members of the KATH HPT clinic, who did not participate in the earlier quantitative study phase and did not consent were excluded. This was done to maintain consistency and continuity in data collection. Those who did not consent were excluded in order not to violate ethical standards and respect for participants' autonomy.” See page 4 lines 187-198 for details.
- The in-depth interviews are well-described, but additional details about how the interview guide was developed (e.g., based on the Chronic Care Model or prior literature) would be beneficial. An appendix with key interview questions or prompts could enhance the manuscript’s completeness.
Response: Thank you for your valuable suggestions. We have incorporated your suggestions by adding further details on how the interview guide was developed with the interview guide added as an appendix. The relevant addition addressing your suggestion now reads, “The interview guide was developed based on the findings of the quantitative data developed from previous literature [10]. The interview guide has been attached as an appendix (Appendix 1). See page 4 lines for details.
3.In qualitative studies, demonstrating credibility, dependability, confirmability, and transferability is crucial. Incorporating information on how you managed reflexivity (e.g., researcher positionality) or how you triangulated data (if at all) will further validate your findings. Providing more on inter-coder reliability or member-checking procedures would also increase confidence in the results.
Response: Thank you for your suggestions. We have incorporated your suggestions by creating a subheading to address your observation and to incorporate your suggestion to read as follows.
“Strategies delineated in establishing trustworthiness were adopted [49]. These included credibility, dependability, confirmability, and transferability [50]. The researchers employed varied strategies to ensure credibility. One of which was an extended engagement with the participants over a period of time in the field offering us an opportunity to engage the participants on their HPT condition from the patients’ self-management practices to social and environmental conditions impacting on their HPT management granting the team a comprehensive insight [51]. This extended engagement gave us the opportunity to build trust and rapport with the participants over time giving us the chance to gain nuanced insights into their beliefs, experiences, and their HPT self-management behaviours, which would not be possible during brief interactions [52]. Bracketing techniques were used in planning, conducting, and reporting this research to identify and address personal ideological assumptions held by the first author and potential influences on this study [53]. The research team was in constant contact throughout all stages to maintain transparency and critically examine the effect of any such assumptions. This brought clarity during data collection and its analysis, improved the quality of decision-making, and finally strengthened the reliability of the findings [54]. Dependability was ensured through the study’s detailed study protocol, ensuring an audit trail of all data collection procedures, and coding precisely. Coding was checked for consistency by the research team members independently [55]. Confirmability was assured through daily debriefing among the research team [52]. Engaging a healthcare qualitative researcher who exerts supervisory checks to review interpretations and findings minimize researcher bias and getting feedback from the supervisory team helped to validate the interpretations and minimizes personal biases brought alternative perspectives [52]. The study ensured transferability through an elaborate description of its methodology, which enables other researchers to replicate it. It described well the setting where the study was carried out, the criteria for including participants, and its convenient sampling approach. These methodological details make the study more transferable, as they guarantee its use in other settings [52].
The study also adopted reflexivity strategies to help minimise the influence of re-searcher assumptions, biases and influences in the research process and findings [56]. The processes adopted consist of a reflexive journal, team debriefings, critical self-analysis, participant validation, seeking an inclusive point of view, and ethical reflexivity. On keeping a reflexive journal: the research team logged our feelings, thoughts, and decisions, which are discussed in an open manner to address any personal or collaborative biases [57]. Daily team debriefing sessions were held not only to discuss operational issues but also reflective logs of the research team regarding interaction with participants and how they influenced the study, both real and assumed. Practices such as these heed recommendations by Berger [58] towards developing collaborative reflexivity among researchers. Also, ethical reflexivity was achieved in conducting and observing international ethical frameworks where participants' autonomy and sensitivities were guaranteed [59]. Finally, seeking inclusive points of view through verification of themes and subthemes with the team tends to triangulate multiple viewpoints to eliminate individual researcher bias. See page 6 lines 254-294 for details.
- While the paper outlines an inductive thematic approach, a more detailed step-by-step description of the coding and theme development process would be helpful. For instance, clarify how the research team resolved disagreements about coding or theme labels, and describe how themes were refined to best represent the data.
Response: We appreciate your insightful observations and suggestions. We have provided a detailed step-by-step description explaining how the inductive thematic approach was employed. The revised description now reads,
“Following the interview process, the Twi-language interviews were translated and transcribed into English by the RA, who is a native Twi speaker and has conducted simi-lar cross-language qualitative research at both undergraduate and Master of Philosophy levels [44]. The first author (F.S) cross-referenced these transcripts with the original audio recordings to ensure accuracy. Data analysis was conducted using NVivo version 12 (QSR International, Ltd., Daresbury Cheshire, UK) through an inductive thematic framework [41, 45, 46]. Next, the transcripts were meticulously examined to identify key themes and points, with significant elements highlighted by the first author (F.S). The research team met regularly to discuss divergent interpretations of data. Collaborative discussions with co-authors K.M.-R, A.S, and T.I.E facilitated the refinement of the analysis framework and thematic categorisation. The first author initiated the grouping of codes into overarching themes, which were subsequently categorised into thematic networks by the research team. The research team, FS, K.M-R, A.S., and T.I.E checked the themes for coherence to ensure that each theme described the main patterns of data. This process involved further analysis to explore relationships among the themes, leading to the identification of pertinent quotes that exemplified the themes and supported the findings. Where disagreement occurred, this was worked through by dialog, returning to the raw data, and ensuring that accounts presented were consistent with participants' narratives. Where necessary, K.M-R, who is an expert in health qualitative research was consulted to offer an impartial perspective. Sub-themes were generated where necessary to capture the subtleties of participant experiences. The final themes were checked against the dataset to ensure the findings were adequately and accurately represented. The team collectively agreed upon the selection of quotes, ensuring they accurately represented the themes derived from the data. Ultimately, the final themes were substantiated with verbatim statements, augmented by relevant demographic details of participants, including sex, age, and HPT status. In summary, the methodological rigour of this study was upheld through a systematic approach to ethics approval, participant recruitment, data collection, and analysis, enabling a nuanced exploration of the enablers and barriers to HPT management in the A/R of Ghana. See page 5 lines 215-248 for details.
- The manuscript mentions multiple factors impacting hypertension management. It might be instructive to discuss how these factors intersect or reinforce each other. A brief section on the interactions between economic burden and knowledge deficits, or how social support can mitigate economic barriers, would enrich the discussion.
Response: Thank you for your suggestion. We have incorporated your suggestion into the discussion and have strengthened the necessary sections appropriately. One of such sections now reads, “Other social interaction factors can be leveraged upon to attenuate the economic burden of patients on their HPT management through strong social support systems such as family, friends, and community-based organizations providing financial help, encouragement, and reminders to patients to take their medication. Besides those mentioned above, the health care setting also supports provider relations that are important in over-coming barriers, offering flexible treatment plans, counselling about financials, and improvement in communication between patients and providers.” See page 15 lines 688-695 for details.
- The findings that highlight limitations in Ghana’s National Health Insurance Scheme (NHIS) and broader health policies are important. Strengthening this section with concrete, evidence-based recommendations—such as specific policy reforms or scalable community interventions—would add real-world applicability to your results.
Response: We appreciate your valuable suggestion. We have amended the section to now read,
“Evidence-based reforms are hence necessary to deal with such challenges. First, extending NHIS coverage to include comprehensive HPT services, including routine screening and essential medications, will reduce out-of-pocket expenses. This has been seen to work in other LMICs, for instance, Rwanda, where universal health coverage improved the management of chronic diseases [83]. The second reduction of delay will also occur because reimbursement procedures through digitalized payment systems will facilitate health facilities in general—a feature piloted in Kenya [84] and ensured by community-based interventions using trained community health workers (CHWs) for HPT screening and management, with established gaps in NCDs across the SSA countries [85,86]. Finally, the integration of HPT management into primary health care and adoption of multisectoral approaches to address the social determinants of health, as witnessed in Thailand's NCD programmes [87] [, might further bolster Ghana's health policy framework.” See page 17 lines 819-831 for details.
- The references to other African or LMIC studies are informative but expanding on how your findings align or diverge from international literature would demonstrate the study’s broader relevance. Highlighting unique Ghanaian contextual factors vs. more universal barriers and enablers could solidify the study’s global health significance.
Response: Thank you for your suggestion. Some revisions have been affected to some relevant portions of the discussion to strengthen the discussion. Some of the new renditions incorporated now reads, “The consensus in the findings explains some methodological convergence across the studies compared and the far-reaching impact on HPT management. However, a major contextual factor in Ghana is the partial coverage of HPT care by the NHIS. Many participants in this study expressed concern that NHIS does not fully cover important aspects of HPT management, hence bearing huge financial burdens. This contrasts with some international models where comprehensive public insurance programmes reduce financial barriers to chronic disease care [67]. For instance, in high-income countries with universal health insurance coverage, like Canada and the United Kingdom, antihypertensive medications and regular monitoring are fully covered under the health plans, which greatly helps in facilitating adherence to the treatment regimens [67]. While financial constraints are usual for most LMICs, Ghana's economic burden is further increased by the structural inefficiencies within the NHIS. Studies from other African nations such as Nigeria and Kenya bring out similar issues where national health insurance schemes provide either partial or inconsistent coverage for chronic conditions and hence their effects on the long-term management of diseases remain limited [68,69]. However, Ghana's NHIS has been commended for being more accessible, in terms of coverage, compared to most LMICs; the scope for change through policy intervention in terms of coverage extension for HPT-related care will, therefore, be very critical [66]. Policymakers must consider strategies to subsidise medication costs and enhance access to affordable healthcare services, including negotiating lower drug prices or establishing community health initiatives focused on delivering affordable care.” See page 14-15 lines 668-688 for details.

Round 2
Reviewer 1 Report
Comments and Suggestions for Authors
Thank you for revising the manuscript based on my suggestions. I am pleased with the changes and recommend it for publication.